# Comparative Study of Lycopene-Loaded Niosomes Prepared by Microfluidic and Thin-Film Hydration Techniques for UVB Protection and Anti-Hyperpigmentation Activity

**DOI:** 10.3390/ijms252111717

**Published:** 2024-10-31

**Authors:** Nattawadee Kanpipit, Sakornchon Mattariganont, Pattanaphong Janphuang, Jureewan Rongsak, Sakda Daduang, Yaowared Chulikhit, Suthasinee Thapphasaraphong

**Affiliations:** 1Faculty of Pharmaceutical Sciences, Khon Kaen University, Khon Kaen 40002, Thailand; natawadee.k@kkumail.com (N.K.); kornmattariganont@kkumail.com (S.M.); 2Synchrotron Light Research Institute (Public Organization), Nakhon Ratchasima 30000, Thailand; pattanaphong@slri.or.th; 3MasterSci Research Co., Ltd., 99 129 Bang Prok, Mueang Pathum Thani District, Pathum Thani 12000, Thailand; jureewan.msr@gmail.com; 4Department of Pharmacognosy and Toxicology, Faculty of Pharmaceutical Sciences, Khon Kaen University, Khon Kaen 40002, Thailand; sakdad@kku.ac.th; 5Department of Pharmaceutical Chemistry, Faculty of Pharmaceutical Sciences, Khon Kaen University, Khon Kaen 40002, Thailand; yaosum@kku.ac.th

**Keywords:** lycopene extract, lycopene-entrapped niosome, thin-film hydration method, microfluidic method, UVB protection, anti-hyperpigmentation

## Abstract

Niosomes are employed for their improved physical properties and stability and as a controlled delivery system. However, their large-scale production and different preparation methods affect their physical properties. The microfluidic method represents a novel approach to the preparation of niosomes that enables precise control and decreases the preparation time and steps compared to alternative methods. The UVB protection and anti-hyperpigmentation activities of lycopene-loaded niosomes prepared by microfluidic (MF) and novel conventional thin-film hydration (THF) methods were compared. Extract powders from tomatoes (T), carrots (C), and mixed red vegetables (MR) were utilized to prepare lycopene-rich extract-entrapped niosomes. The resulting niosome formulations were characterized by particle size, polydispersity index (PDI), zeta potential, FT-IR spectra, entrapment efficiency, lycopene-release profile, permeation, and stability. The lycopene extract–niosome formulations were evaluated for their potential to provide UVB protection to human keratinocytes (HaCaT) and for their anti-melanogenesis effects on B16F10 melanoma cells. The results indicated that niosomes prepared by the MF method exhibited high uniformity and homogeneity (reflected by a low PDI value) and maintained smaller sizes when processed through a chip utilizing a hydrodynamic flow-focusing (HFF) platform compared to THF niosomes. The release kinetics of all lycopene–niosome formulations followed the Korsmeyer–Peppas model. The FT-IR spectra indicated that lycopene was incorporated into the niosome bilaminar membrane. Moreover, niosomes obtained from MF demonstrated enhanced stability during heating–cooling cycles, along with high UVB protection and anti-melanogenesis effects. Therefore, these developed niosome preparation methods could be effectively applied to topical products.

## 1. Introduction

Lycopene is an important carotenoid pigment found in various vegetables and fruits that offer health benefits [1]. It has a non-conjugated double-bond structure and 11 conjugated acyclic structures that confer strong antioxidant properties [2]. Lycopene provides protection against ultraviolet (UV) radiation [3] and exhibits inhibitory activities against elastase and tyrosinase [4]. It is abundant in mature red plant fruits such as tomatoes, papayas, pink grapefruits, pink guavas, and watermelons [5]. Natural lycopene is plentiful and is used in health product supplements [6]. These red fruits, especially tomatoes, are also rich sources of vitamins, minerals, fiber, protein, essential amino acids, monounsaturated fatty acids, phytosterols, and other bioactive phenolic compounds (including quercetin, kaempferol, naringenin, and lutein, as well as caffeic, ferulic, and chlorogenic acids) and anthocyanin compounds. These bioactive compounds possess potential antioxidant, anticancer, anti-inflammatory, antidiabetic, antithrombotic, and antimicrobial effects as well as preventative effects against neurodegenerative and cardiovascular diseases [7]. Furthermore, lycopene-rich products and polyphenol combinations have been shown to provide protection against UVB irradiation by inhibiting NF-κB [8] and UVB-induced ornithine decarboxylase (ODC) activity, and by preventing inflammatory responses [9]. Previous in vitro studies have reported that tomato extracts containing lycopene also inhibit tyrosinase (73.12%), indicating a high potential for use as a cosmetic ingredient, particularly as an antiaging agent [4]. Carotenoids have been shown to protect against melanogenic intermediates and exogenous DNA damage [10], and a previous study reported that carrot extract has a protective effect on the skin (fibroblasts) and prevents apoptosis of fibroblast cells [11]. However, there are currently no studies investigating the UVB protection and anti-melanogenesis properties of mixed red vegetables. Lycopene has several limitations to overcome before it can be developed as a cosmetic ingredient. Lycopene can degrade through isomerization and oxidation when exposed to various heating and light conditions [12] and, since lycopene is a non-polar compound, it is insoluble in water. Moreover, direct application of lycopene to the skin can lead to skin irritation, cytotoxicity, and poor absorption [13]. Therefore, the current study aimed to develop niosomes that enhance skin absorption of lycopene while mitigating skin irritation and toxic side effects.

Nanotechnology offers promising approaches for developing nanoemulsions, liposomes, and niosomes to improve the stability, permeability, and targeted delivery of bioactive compounds. These nanoparticles can help maintain effective concentrations of active compounds at the site of application, reducing the need for frequent administration [14].

Niosomes are a vesicular delivery system composed of a mixture of a non-ionic surfactant and cholesterol that forms a bilayer structure. They can encapsulate hydrophobic compounds within the bilayer and hydrophilic compounds within the aqueous core [15]. They can also be made smaller depending on the specific delivery requirements needed to produce nano-carrier systems. Niosomes are employed for their improved physical properties and stability, and as a controlled delivery system, which requires their formulation to be biocompatible to reduce toxicity and adverse effects [16]. Niosomes allow for the prolonged release of active ingredients with moisturizing effects on the skin and have been utilized in products with anti-aging, whitening, and antioxidant effects, and both pharmaceutical and cosmeceutical applications [17]. Several studies have explored the biomedical applications of lycopene-loaded niosomes.

One lycopene niosome formulation prepared by the adsorption hydration method demonstrated anti-cancer activity against MCF-7 and HeLa cell lines, with enhanced entrapment efficiency [1]. Another lycopene noisome formulation prepared by the adsorption hydration method showed anti-diabetic activity by reducing blood glucose levels [18]. Moreover, a previous study reported the incorporation of lycopene into cationic niosomes by the reverse phase evaporation method to deliver genetic material to the rat retina for the treatment of inherited retinal diseases [19]. However, there are no studies to date investigating lycopene–niosome formulations for cosmetic applications, UVB protection, and anti-hyperpigmentation.

Various methods are employed for preparing niosomes, depending on the desired vesicle size, size distribution, entrapment efficiency, and number of double-layer formulations required. These include thin-film hydration, sonication, reverse-phase evaporation, and ether injection. The conventional thin-film hydration method involves dissolving surfactant and cholesterol in an organic solvent, which is then evaporated to produce dried films. Subsequently, rehydration is achieved by adding an aqueous phase. The obtained niosomes can undergo sonication to produce smaller vesicles with a more homogenous size distribution [20].

The microfluidic method represents a novel approach to the preparation of niosomes that enables precise control of the resulting particle size via a chip that regulates the flow of the organic and aqueous phases during mixing. During the mixing process, two or more inlet streams containing either lipids dissolved in an organic solvent or an aqueous phase are mixed and microfluidic equipment is employed to achieve the desired niosome formulation [21]. Microfluidic techniques can significantly decrease preparation time and steps compared to alternative methods [1].

Lycopene–niosomes prepared using Span 60 and cholesterol via the adsorption–hydration method have shown enhanced entrapment and improved lycopene stability as well as prolonged release of lycopene [22]. However, this technique requires many steps, making it unsuitable for industrial scale. Furthermore, there are currently no reports comparing the preparation of lycopene-loaded niosomes from various vegetable-source-rich carotenoids using microfluidic methods with traditional thin-film methods.

This study aims to evaluate the impact of different preparation methods on the characterization of lycopene-extract niosomes derived from vegetables, as well as their biological activities. Niosomes containing extracts from tomatoes (T), carrots (C), and mixed red vegetables (MR) were prepared using both microfluidic and thin-film methods. Key parameters such as particle size, polydispersity index (PDI), encapsulation efficiency (%), morphology (as observed under a transmission electron microscope (TEM)), and lycopene-release profiles were analyzed. Additionally, the in vitro biological activities of these formulations, including UVB protection and anti-hyperpigmentation effects on human skin cells, were assessed.

## 2. Results and Discussion

### 2.1. Determination of Lycopene Content in Vegetables

Lycopene content in tomato (T), carrot (C), and mixed red vegetable (MR) powders was determined by UV spectrophotometry. Scanning across wavelengths from 200 to 600 nm (Figure 1) revealed prominent peaks at 442, 475, and 503 nm corresponding to the standard lycopene spectrum in n-hexane (Figure 1A). The total lycopene content was calculated using the Beer–Lambert law at 475 and 503 nm, yielding a mean absorption coefficient (A1^%^_1cm_) of 3450 using the mean molecular weight of lycopene, 536.9 g/mol [23]. The lycopene contents in tomato, carrot, and mixed red vegetables were found to be 165.65 ± 5.78, 23.17 ± 0.60, and 91.64 ± 25.65 mg/100 g, respectively, as shown in Figure 1B. This UV spectrophotometry method is a fast and inexpensive approach for the quantitative determination of lycopene in vegetable powders [1].

### 2.2. Physical Characterization of Niosome

#### 2.2.1. Particle Size, Polydispersity (PDI), Zeta Potential, and Entrapment Efficiency

The particle size, polydispersity (PDI), zeta potential, and entrapment efficiency of niosome formulations prepared by thin-film hydration (THF) and microfluidic (MF) methods are summarized in Table 1. The particle size of niosomes ranged from 237.97 to 457.47 nm. Niosome formulations produced using the MF method generally resulted in significantly smaller particle sizes (237.97–281.73 nm) compared to those produced with the THF method (245.17–457.47 nm). The PDI values of all formulations were less than 1.00 and the PDI values of niosome formulations produced by the MF method were significantly lower than those produced by the THF method. The zeta potential measurements for all formulations were below −30 mV and the zeta potentials of the BN−M, TN−M, and CN−M niosome formulations prepared by the MF method were more negative than their THF counterparts shown in the zeta potential graphs in Appendix A. Both the MF and THF methods produced niosomes with lycopene entrapment efficiencies of more than 90%.

The MF method enables precise control over niosome properties by adjusting the flow rate ratio of the organic and aqueous phases within microchannels. This makes it a promising technique for developing and optimizing nanodelivery systems with controlled size and polydispersity, which are important in biotechnological applications [24]. High lycopene encapsulation efficiency across all formulations was achieved due to lycopene’s hydrophobic nature and its interaction with the hydrophobic regions of the niosome bilayer membrane. Non-ionic surfactants with a hydrophilic–lipophilic balance (HLB) value of 4.3 are suitable for forming niosomes with lycopene compounds. Cholesterol helped to stabilize the vesicles by interacting with the surfactant’s hydrophobic alkyl end, ensuring efficient lycopene encapsulation [25]. The observed decrease in particle size and narrow distribution (low PDI value) in MF-produced niosomes aligns with previous findings [26].

#### 2.2.2. In Vitro Lycopene-Release Study

The lycopene-release profiles of six niosome formulations (TN−M, TN−T, CN−M, CN−T, MRN−M, and MRN−T) and their respective extract solutions (TS, CS, and MRS) are shown in Figure 2 and Appendix A. Lycopene release from the niosomes was sustained, with cumulative release peaking within 24 h. Tomato–niosome formulations (TN−M and TN−T) exhibited maximum cumulative releases of 81.75% and 76.08%, respectively, while carrot–niosome formulations (CN−M and CN−T) reached 77.98% and 76.96%, respectively. Mixed red vegetable–niosome formulations (MRN−M and MRN−T) showed slightly lower cumulative releases of 65.75% and 62.97%, respectively. The release profiles from both MF-prepared and THF-prepared niosomes were comparable, and all formulations demonstrated prolonged lycopene release.

A previous study reported that the sustained release of lycopene from niosomes was due to the affinity of niosomes for phospholipids, which facilitated absorption and enabled slower release. In that study, there was a rapid initial release of lycopene within 10 h, followed by a slower continuous release until 72 h [27].

In the current study, kinetic analysis revealed that lycopene release from all niosome formulations followed the Korsmeyer–Peppas model (*n* less than 0.45, in Appendix A) with a Fickian diffusion (Case I diffusional) mechanism. This suggests that lycopene was retained in the niosome bilayer and its slow release was primarily controlled by diffusion from the outer surface. This confirms a previous report that showed that the Korsmeyer–Peppas model had the best fit for lycopene–niosomes prepared by an adsorption–hydration method [27]. Lycopene’s sustained release is essential for maintaining optimal concentrations for biological activity while reducing administration frequency. Microfluidic mixing enabled particle self-assembly and drug loading in a single step, eliminating the need for post-preparation size reduction [28].

#### 2.2.3. Transmission Electron Microscopy (TEM) Measurements

TEM micrographs of niosome formulations prepared by the THF and MF methods revealed spherical unilamellar structures with particle sizes ranging from 245.17 to 457.47 nm for the THF method, and 237.97 to 281.73 nm for the MF method (Figure 3 and Figure 4). The morphology of spherical vesicles surrounded by a thin layer was consistent with a previous report of lycopene–niosomes [29].

#### 2.2.4. Stability Evaluation

Niosome formulations from both methods were subjected to stability evaluation under heating–cooling storage conditions. The zeta potential (mV), particle size (nm), and PDI values showed no significant differences (*p* > 0.05) for all formulations from both methods before and after storage conditions (Table 2). However, the entrapment efficiency (%) showed significant differences (*p* < 0.05) for the TN, CN, and MRN formulations prepared by the thin-film method, but no significant differences (*p* > 0.05) for those prepared by the microfluidic method. This result indicates that formulations prepared using the microfluidic method can effectively replace those from the conventional thin-film methods while maintaining high stability. Lycopene from various fruits and vegetables (tomatoes, carrots, and mixed red vegetables) loaded into niosomes showed enhanced stability due to its incorporation into the outer wall of the niosomes. Previous research has shown that niosomes protect lycopene by embedding it within their bilayer structure, thereby shielding it from oxidative stress and light exposure while maintaining the formulation’s stability [27].

#### 2.2.5. FT-IR Spectroscopy

The FT-IR spectrum of lycopene (Appendix A) shows functional groups and wavenumbers in the 400–4000 cm^−1^ range that were identified as C−H stretching (2920.10 cm^−1^), CH_2_ symmetric stretching (2849.74 cm^−1^), C = C stretching (1539.63, 1559.52, and 1575.27 cm^−1^), CH_2_ bending (1465.50 cm^−1^), C−C stretching (1075.30 cm^−1^), and C = C bending (834.83 cm^−1^).

The FT-IR spectra of lycopene extracts are shown in Appendix A; they reveal peaks of OH stretching (3311.58, 3318.88, 3284.75 cm^−1^, respectively), CH stretching (2800–3000 cm^−1^), CH_2_ bending (1376.56, 1375.83, and 1335.50 cm^−1^), and C−C stretching (1028.55, 1026.63, and 1016.42 cm^−1^) in tomato, carrot, and mixed red vegetables, respectively. While all samples exhibited similar FT-IR spectral patterns, the mixed red vegetable spectrum displayed more distinct wavenumber positions compared to the others.

The FT-IR spectrum of cholesterol (Appendix A) showed O−H stretching (3401.15 cm^−1^), C−H stretching (2800–3000 cm^−1^), CH_2_ bending (1462.77 cm^−1^), and C−O stretching (1053.56 cm^−1^). The FT-IR spectrum of Span 60 (Appendix A) showed O−H stretching (3379.33 cm^−1^), C−H stretching (2916.29/2849.24 cm^−1^), C = O strong ester bonding (1735.46 cm^−1^), CH₂ bending (1466.95 cm^−1^), C−O stretching (1173.91 cm^−1^), and CH_2_ rocking vibrations (720.82 cm^−1^).

The FT-IR spectra of niosomes prepared by the thin-film hydration method presented similar patterns, as shown in Appendix A, for blank niosomes (BN−T), tomato niosomes (TN−T), carrot niosomes (CN−T), and mixed red vegetable niosomes (MRN−T), respectively. However, shifts in wavenumber from the BN−T baseline were greater for TN−T and CN−T than they were for MRN−T. The C−O stretching peak at 1038.39 cm^−1^ for BN−T shifted to 1025.93 cm^−1^ for TN−T and to 1027.88 cm^−1^ for CN−T.

Niosomes prepared by the microfluidic method also presented similar FT-IR patterns, as shown in Appendix A, for blank niosomes (BN−M), tomato niosomes (TN−M), carrot niosomes (CN−M), and mixed red vegetable niosomes (MRN−M). Once again, TN−M and CN−M niosomes showed a greater shift in wavenumber than MRN−M compared to BN−T. The C−O stretching peak at 1038.47 cm^−1^ for BN−M shifted to 1026.47 cm^−1^ for TN−M and to 1028.88 cm^−1^ for CN−M. In Figure 5, the peaks of blank niosomes prepared by both methods are sharper compared to any of the lycopene-entrapped niosomes. The FT-IR spectra of niosomes containing lycopene extracts showed a similar peak pattern of interaction between lycopene, Span 60, and cholesterol via hydrogen bonding [27]. Thus, lycopene is predicted to be embedded within the bilayer of surfactant and cholesterol in the niosome structure through hydrogen bonding, as shown in Figure 6.

#### 2.2.6. Biological Properties of Niosome Formulations

##### UVB Protection Effects 

The cytotoxicity of extracts and niosome formulations against HaCaT cells was evaluated using the MTT assay. HaCaT cells were pretreated for 24 h with various concentrations of tomato (TS), carrot (CS), and mixed red vegetable (MRS) extracts, ascorbic acid (AA), and their niosome formulations (BN−T, TN−T, CN−T, MRN−T prepared by the thin-film hydration method and BN−M, TN−M, CN−M, MRN−M prepared by the microfluidic method). All samples and niosome formulations were not cytotoxic to HaCaT cells under these conditions, as indicated by cell viability values greater than 80% (Appendix A).

The UVB protective effects were evaluated by exposing HaCaT cells to a 10 mJ/cm^2^ dose of UVB radiation following pretreatment for 24 h with tomato (TS), carrot (CS), and mixed red vegetable extracts (MRS), as well as niosome formulations prepared by the thin-film hydration method (BN−T, TN−T, CN−T, MRN−T) and the microfluidic method (BN−M, TN−M, CN−M, MRN−M), as shown in Figure 7. The 10 mJ/cm^2^ dose of UVB radiation caused significant toxicity (*p* > 0.05) compared to non-irradiated cells ((−) control). Pretreatment with extracts and niosome formulations provided significant UVB protection to HaCaT cells (*p* < 0.05) compared to untreated, UVB irradiated cells (UVB). Blank niosomes prepared using both methods (BN−T and BN−M) did not provide UVB protection, as shown in Figure 8. The extract-loaded niosomes (TN−T, CN−T, MRN−T, TN−M, CN−M, MRN−M) showed slightly greater UVB protection than extract solutions alone. However, all niosome formulations offered less UVB protection compared to the positive control, ascorbic acid (AA), and the negative control ((−) control).

This result suggests that niosome formulations from both preparation techniques (microfluidic and thin-film hydration methods) could provide comparable UVB protection due to their high entrapment efficiencies and enhanced lycopene stability. According to previous studies, lycopene is well known for its antioxidant properties, which can protect against acute photodamage induced by UVB exposure [9]. Figure 9 presents the cell morphology of HaCaT cells after 24 h, comparing non-treated cells (Figure 9A) with those exposed to 10 mJ/cm^2^ UVB irradiation (Figure 9B) and those pre-treated with tomato, carrot, and red vegetable solutions (Figure 9D–F) as well as niosome formulations prepared by the microfluidic method (BN−M, TN−M, CN−M, MRN−M) (Figure 9H–J). The results show that cells treated with the extracts and niosomes exhibited UVB protection.

Moreover, 6-diamidino-2-phenylindole (DAPI) staining was used to investigate the protective effect of lycopene niosome on HaCaT cells, as shown in Figure 8. The fluorescent images revealed that non-treated cells (Figure 8A) had intact nuclei, which increased the intensity of nuclei, while cells exposed to 10 mJ/cm^2^ UVB irradiation (Figure 8B) displayed a decreased frequency of nuclei. Moreover, cells pretreated with positive control (ascorbic acid), tomato, carrot, and red vegetable solutions, as well as niosome formulations prepared by the microfluidic method (BN−M, TN−M, CN−M, and MRN−M) showed an increased frequency of nuclei (Figure 8C–J). These results confirmed that niosomes exhibited UVB protection.

##### Anti-Melanogenic Effects

Cytotoxicity against B16F10 cells pretreated for 48 h with tomato (TS), carrot (CS), and mixed red vegetable extracts (MRS) and their niosome formulations prepared by the thin-film hydration method (BN−T, TN−T, CN−T, MRN−T) and the microfluidic method (BN−M, TN−M, CN−M, MRN−M) was assessed using the MTT assay. All niosome formulations containing 10% (*w*/*v*) extracts showed no toxic effects (Appendix A). All formulations were further investigated for tyrosinase inhibition and melanin content.

B16F10 cells were pretreated with tomato (TS), carrot (CS), and mixed red vegetable extracts (MRS), and their niosome formulations prepared by the thin-film method (BN−T, TN−T, CN−T, MRN−T) and the microfluidic method (BN−M, TN−M, CN−M, MRN−M) for 48 h and stimulated with α-melanocyte stimulating hormone (α-MSH) to investigate melanogenesis.

For tyrosinase inhibition, pretreatment with all extracts and niosome formulations provided significantly higher tyrosinase inhibition in B16F10 cells (*p* < 0.05) compared to the control (with α-MSH), as shown in Figure 10A. Additionally, blank niosomes from both methods (BN−T and BN−M) exhibited low tyrosinase inhibition, while MRN−M demonstrated the highest tyrosinase inhibition, comparable to the positive control (KA).

For melanin content, the control (with α−MSH) resulted in a significantly higher melanin content (*p* > 0.05) compared to the negative control (without α-MSH), as shown in Figure 10B. Pretreatment with extracts from tomato (TS), carrot (CS), and mixed red vegetable solutions (MRS), as well as niosome formulations (CN−T, MRN−T, TN−M, CN−M, MRN−M), showed reduced melanin content. However, cells treated with niosome formulations prepared by the microfluidic method (TN−M, CN−M, MRN−M) resulted in significantly lower melanin content compared to cells treated with extract solutions and niosome formulations prepared by the thin-film hydration method (TN−T, CN−T, MRN−T).

These results suggest that niosome formulations from both preparation techniques demonstrated an enhanced reduction in melanin content and tyrosinase inhibition due to their high entrapment efficiencies.

Tyrosinase is important for melanin synthesis by melanocytes in the epidermis, and α-MSH is a precursor and inducer of melanin pigmentation activation and is used as a marker for determining melanin content [30]. The lower melanin contents after treatment with niosomes prepared by the microfluidic method suggest that the fewer formulation steps in the microfluidic method may reduce lycopene degradation. Niosome formulations prepared by the microfluidic method exhibit improved stability and biological activity [31]. According to a previous study, lycopene from tomatoes showed in vitro tyrosinase inhibition (41.16% ± 5.41%) [4]. However, lycopene compounds have not yet been reported to inhibit intracellular tyrosinase or melanin formation in B16F10 cells.

## 3. Materials and Methods

### 3.1. Materials

Extract powders of tomato (T), carrot (C), and mixed red vegetables (MR) were obtained from MasterSciResearch Co. Ltd., Mueang Pathum Thani District, Thailand (Appendix A). Potassium dihydrogen phosphate (KH_2_HPO_4_), sodium hydrogen carbonate (NaHCO_3_), disodium hydrogen phosphate (Na_2_HPO_4_), sodium chloride (NaCl), and sodium carbonate (Na_2_CO_3_) were obtained from Ajax Finechem Pty Limited (Wollongong, NSW, Australia). Ascorbic acid, L-DOPA, α-MSH, and tyrosinase enzyme (≥2000 units/mg solid) were obtained from Sigma-Aldrich (St. Louis, MO, USA). Phosphate-buffered saline (PBS), fetal bovine serum (FBS), penicillin–streptomycin (10,000 U/mL), trypsin EDTA, and Dulbecco’s modified Eagle’s medium (DMEM) were acquired from Gibco (Gaithersburg, MD, USA). 3-[4,5-dimethylthiazol-2-yl]-2,5-diphenyltetrazolium bromide (MTT) and dimethyl sulfoxide (DMSO) were supplied by Thermo Fisher Scientific, (Waltham, MA, USA). The human keratinocyte cell line (HaCaT) and murine melanoma cell line (B16F10) were purchased from the American Type Culture Collection, Manassas, VA, USA. A microfluidic chip featuring a cross-junction microchannel design with a diameter of 0.5 mm was acquired from the Synchrotron Light Research Institute (SLRI), Thailand.

### 3.2. Determination of Lycopene Content

Lycopene content was analyzed using UV-VIS spectrophotometry based on the Beer–Lambert law. The samples were prepared at concentrations of up to 10% (*w*/*v*) in distilled water. Lycopene was extracted from the solutions with n-hexane, followed by vortexing and stirring for 2 h. The absorption spectrum (400 to 600 nm) was recorded using UV-VIS spectrophotometry (Thermo Scientific: GENESYS 50, Waltham, MA, USA). Lycopene content was quantified using the Beer–Lambert equation [32]. Moreover, the determination of lycopene content analyzed by the UV-VIS spectrophotometry method was comparable to the results obtained using the validated HPLC method [33]. The information for HPLC method validation and lycopene content are presented in Appendix A.
A = εbc (1)

A = Absorbance at the maximum wavelength of lycopene

ε = Molar absorptivity (L mol^−1^ cm^−1^)

c = Concentration of lycopene (mol L^−1^)

b = The path length (cm)

### 3.3. Preparation of Niosomes

#### 3.3.1. Conventional Thin-Film Hydration (THF) Method

Span 60 and cholesterol were combined in a 1:1 molar ratio and dissolved in ethanol to create the organic phase. Blank niosomes (BN−T) and lycopene–niosomes from tomato (TN−T), carrot (CN−T), and mixed red vegetable (MRN−T) were formulated as shown in Table 3. The organic phase was evaporated under reduced pressure in a rotary evaporator to produce a thin film. The film was dried for 30 min and then hydrated with 10 mL of 10% extract in PBS (pH 7.4) for 30 min under rotation to produce niosome suspensions. The suspensions were sonicated twice: 5 min in a bath sonicator (CREST 230 T, NY, USA) followed by 10 min of sonication using a probe sonicator (Vibra-Cell™, VCX-500, Newtown, CT, USA) (40% amplitude, 10 s pulse with a 10 s pause) to obtain the niosome formulations shown in Appendix A [34]. 

#### 3.3.2. Microfluidic Mixing (MF) Method

Niosomes were prepared using microfluidic mixing on a lab-on-a-chip platform with a hydrodynamic flow-focusing (HFF) setup. The microfluidic chip featured a cross-junction microchannel micromixer with a 0.5 mm diameter, incorporating three inlets and one outlet, as shown in Figure 11. The organic phase consisted of the non-ionic surfactant Span 60 (25.80 mg) and cholesterol (22.80 mg) in 10 mL of ethanol. The aqueous phase was prepared by dissolving 200 mg of sample powder in 10 mL of PBS (10 mM, pH 7.4). The aqueous and organic phases were pumped into the microfluidic device by a peristaltic pump (Multi-channel peristaltic pump BT100-1L-up to 24 channels, Shaanxi, China) at a controlled flow rate of 0.5 mL/min. Encapsulation was achieved by introducing the aqueous phase before microfluidic mixing with the organic phase. Niosomes were collected from the chamber outlet, and the residual organic solvent was removed via rotary evaporation. The final volume was adjusted to 20 mL to obtain blank niosomes (BN−M) and lycopene–niosomes from tomato (TN−M), carrot (CN−M), and mixed red vegetables (MRN−M), as shown in Appendix A [35]. 

### 3.4. Physical Characterization of Niosomes

#### 3.4.1. Entrapment Efficiency

Lycopene entrapment efficiency was determined using ultrafiltration with Amicon^®^ ultra-4 device (Ultracel-100K, 100 Kda cutoff, Sartorius AG, Göttingen, Germany). Un-entrapped lycopene was separated from entrapped lycopene by centrifugation (Kubota 6200, Tokyo, Japan) at 5000 rpm and 4 °C for 30 min. The concentration of free lycopene was measured using a UV spectrophotometer (Thermo Scientific: GENESYS 50, MA, USA). The entrapment efficiency of lycopene was calculated by subtracting the amount of un-entrapped lycopene from the total lycopene added for niosome preparation as follows [36].
Entrapment efficiency (%) = [(A_total_ − A_u_)/A_total_] × 100(2)
where A_total_ is the total lycopene added and A_u_ is the un-entrapped lycopene.

#### 3.4.2. Vesicle Size Measurement

The particle size and polydispersity index (PDI) of niosomes were measured using dynamic light scattering with a Zetasizer apparatus (Malvern Instruments Ltd., Worcestershire, UK). All experiments were performed in triplicate [37].

#### 3.4.3. Zeta Potential Measurement

Zeta potential, indicating particle stability and electrostatic repulsion, was measured using a Zetasizer Nano ZS-90 (Malvern Instruments Ltd., UK) at 25 °C [38].

#### 3.4.4. Transmission Electron Microscopy (TEM)

Samples were dropped onto carbon-coated copper grids and air-dried at room temperature. The grids were imaged under TEM (FEI, TECNAI G2, Eindhoven, The Netherlands) with an accelerating voltage of 80 kV [39]. 

#### 3.4.5. Stability Study

Niosomes were subjected to thermal stability testing using a heating–cooling cycle (4 °C and 45 °C every 24 h) for 7 days. Particle size, PDI, Zeta potential, and entrapment efficiency were evaluated before and after storage [18].

#### 3.4.6. In Vitro Release of Lycopene

Lycopene release was assessed using a Franz diffusion cell with a dialysis membrane (CelluSep^®^, T4, MWCO 12–14 kDa, Membrane Filtration Products, TX, USA) at 37 °C in phosphate buffer (pH 7.4). One-milliliter aliquots of niosome formulations were added to the donor chamber. Samples were collected from the receptor chamber at 0.5, 1, 2, 4, 6, 8, 12, and 24 h and analyzed for lycopene content using UV spectrophotometry (Thermo Scientific: GENESYS 50, MA, USA) [18].

#### 3.4.7. Kinetic Release Prediction

Lycopene release kinetics were analyzed using (1) zero-order (cumulative percentage of release and time), (2) first-order (log of cumulative percentage of release and time), (3) Higuchi (cumulative percentage of release and square root of time), and (4) Korsmeyer–Peppas (log of cumulative percentage of release and log of time) models [18]. Data were fitted using linear regression and evaluated using the correlation coefficient (r). The Korsmeyer–Peppas model was used to analyze the mechanism of drug release and the diffusion kinetics. The exponent (n) obtained from the slope of linear regression is indicative of the release mechanism where *n* < 0.5 indicates Fickian diffusion, 0.5 < *n* < 1 indicates non-Fickian transport, *n* = 1 indicates Case II (relaxational) transport, and *n* > 1 indicates super case II transport [40].

#### 3.4.8. Fourier Transform Infrared (FT-IR) Spectroscopy

The niosomes were air-dried before measurement. FT-IR in ATR mode was performed using a Bruker Tensor 27 FT-IR spectrometer (Billerica, MA, USA) with an ATR cell to verify the encapsulation of all niosome samples and all extracts. The spectral transmittance across the range of 4000 to 400 cm^−1^ was evaluated [41].

### 3.5. Biological Properties of Niosome Formulations

#### 3.5.1. Anti-Melanogenic Effects

##### Cytotoxicity Assay of B16F10 Cells

The cytotoxicity of niosome formulations in B16F10 cell lines cultured in DMEM supplemented with 10% FBS, 100 µg/L streptomycin, and 100 IU/mL penicillin at 37 °C and 5% CO_2_ was determined. Cells were seeded into 96-well plates at 1 × 10^4^ cells/well and incubated for 24 h at 37 °C under 5% CO_2_. After replacing the medium with niosome formulations to give a total volume of 100 µL in the culture medium, cells were incubated for 48 h at 37 °C under 5% CO_2_ and cell viability was measured using the MTT assay at 570 nm with a microplate reader (EnSight^®^ multimode microplate reader, PerkinElmer, Waltham, MA, USA) [32]. Cell viability was calculated as follows:% Cell viability = (absorbance of sample/absorbance of (−) control) × 100(3)

##### Melanin Measurement [32]

B16F10 cells (1 × 10^5^ cells/well in 12-well plates) were incubated for 24 h at 37 °C under 5% CO_2_. After 24 h, the medium was aspirated and then replaced with each sample or 100 µg/mL of kojic acid (positive control) to give a total volume of 1500 µL in the culture medium. It was stimulated with α-MSH (200 nM) and incubated for 48 h at 37 °C under 5% CO_2_. After 48 h, the cells were rinsed twice with PBS and lysed using a lysis buffer (20 mM sodium phosphate (pH 6.8) and 1% Triton X-100). The cells were centrifuged at 12,000 rpm for 15 min and the supernatant was discarded. The pellets were suspended in 10% DMSO in 1 N NaOH for 1 h at 80 °C to dissolve melanin. The melanin content was measured by absorbance at 475 nm using a microplate reader (EnSight^®^ multimode microplate reader, PerkinElmer, Waltham, MA, USA).

##### Intracellular Tyrosinase Activity Measurement 

The B16F10 cells (1 × 10^5^ cells/well in 12-well plates) were incubated for 24 h at 37 °C under 5% CO_2_. After 24 h, the medium was aspirated and then replaced with each sample or 100 µg/mL of kojic acid (positive control) to give a total volume of 1500 µL in the culture medium. Cells were stimulated with α-MSH (200 nM) and incubated for 48 h at 37 °C under 5% CO_2_. After this time, the cells were washed with cold PBS and lysed with lysis buffer (0.1 M sodium phosphate (pH 6.8) and 0.1% Triton X-100). The cell lysates were clarified by centrifugation at 12,000 rpm for 5 min at 4 °C. Lysates were dissolved in 0.1 M sodium phosphate at pH 6.8. The protein content in the supernatant was determined using the BCA method, in which 100 µg/mL of sample was mixed with 100 µL of 0.1% L-DOPA in PBS at pH 6.8. This was followed by incubation for 20 min at 37 °C. Absorbance was measured at 475 nm with a microplate reader (EnSight^®^ multimode microplate reader, PerkinElmer, Waltham, MA, USA) and calculated as tyrosinase inhibition [32].

#### 3.5.2. Protection of Niosome Formulations Against UVB Radiation

##### Cell Lines and Culture Conditions

Human keratinocyte cell lines (HaCaT) were purchased from the American Type Culture Collection, Manassas, VA, USA. They were cultured in Dulbecco’s modified Eagle’s medium (DMEM) (GIBCO, Paisley, UK) with 10% (*v*/*v*) fetal bovine serum (GIBCO, Paisley, UK) and 1% (*v*/*v*) penicillin/streptomycin (GIBCO, Paisley, UK) at 37 °C under a humidified 5% CO_2_ atmosphere. When the cells reached 70–80% confluence, they were trypsinized with 0.25% trypsin–EDTA according to the parameters of the study [21].

##### Cytotoxicity Assay for HaCaT Cells

HaCaT cells were seeded into 96-well plates at an initial concentration of 1 × 10^4^ cells/well in culture medium and incubated for 24 h at 37 °C under 5% CO_2_. The medium was aspirated and then replaced with each sample to provide a total volume of 100 µL in the culture medium. They were then incubated for 24 h at 37 °C under 5% CO_2_. The medium was removed from the 96-well plates and cell viability was measured with the MTT assay at 570 nm using a microplate reader (EnSight^®^ multimode microplate reader, PerkinElmer, Waltham, MA, USA) [32]. Cell viability was calculated using Equation (3).

##### UVB Irradiation System

The illumination system consisted of a Philips UVB Broadband TL 20 W/12 phototherapy lamp (Philips, Amsterdam, The Netherlands). Irradiance was measured using a photoradiometer (UV 340B). HaCaT cells were maintained at 37 °C under 5% CO_2_ and washed with phosphate-buffered saline, leaving a thin buffer layer for irradiation. The negative control group was the cells that were not exposed to UVB irradiation [32].

##### UVB Protection Effects on HaCaT Cells

Cytotoxicity tests were conducted with UVB irradiation against HaCaT cells using an MTT assay. HaCaT cells were pretreated with each sample for 24 h at 37 °C under 5% CO_2_. Irradiation was conducted at 10 mJ/cm^2^ with a UVB lamp. MTT assays were performed 24 h after UVB irradiation [32].

##### The Morphology of HaCaT Cells Using Fluorescence Microscopy

HaCaT cells were pretreated with each sample for 24 h at 37 °C under 5% CO_2_. Irradiation was conducted at 10 mJ/cm^2^ with a UVB lamp. The morphological changes in the nuclei of cells were determined. The cells were harvested and washed with PBS and fixed using 4%paraformadehyde for 15 min at room temperature. The cells were then stained with 1.0 µg/mL of DAPI solution (Sigma-Aldrich Chemical, St. Louis, MO, USA) for 10 min at room temperature in the dark and washed with PBS. The morphology of the stained cells was evaluated by fluorescence microscopy (Invitrogen EVOS M7000, Thermo Scientific, MA, USA) [34].

### 3.6. Statistical Analyses

Data were analyzed using SPSS Version 28 (SPSS Inc., Chicago, IL, USA; Licensed KKU software). Statistical results are presented as mean ± standard deviation (SD). Differences between groups were assessed using one-way analysis of variance (ANOVA) followed by Dunnett’s test and Duncan’s analysis for post hoc testing. *p*-values less than 0.05 were considered to be significant.

## 4. Conclusions

In this research, we evaluated the effects of various niosome formulations containing lycopene-rich extracts from tomatoes, carrots, and mixed red vegetables, and compared the niosome preparation methods between the conventional thin-film hydration method and microfluidic method. The results showed that the microfluidic method was more efficient than the thin-film hydration method for large-scale production, as it requires fewer steps and offers greater uniformity, homogeneity (lower PDI values), and better control over smaller niosome sizes. These lycopene-loaded niosomes demonstrated sustained release following Korsmeyer–Peppas kinetics, ensuring optimal lycopene concentrations for biological activity while reducing administration frequency. FT-IR analysis confirmed that the controlled release of lycopene in the lycopene–Span 60–cholesterol was via hydrogen bonding. Both production methods improved the stability of lycopene within the niosome formulations under thermal stress conditions. The lycopene–niosomes derived from tomatoes, carrots, and mixed red vegetables also exhibited UVB protective potential, enhancing the viability of HaCaT cells exposed to UVB irradiation. When applied topically, these niosomes showed potential skin-whitening effects by decreasing melanin content and inhibiting tyrosinase activity, with no observed cytotoxicity in B16F10 cells. These findings suggest that lycopene-loaded niosomes could be incorporated in skincare products such as gels or creams, offering both UV protection and skin-lightening benefits.

## Figures and Tables

**Figure 1 ijms-25-11717-f001:**
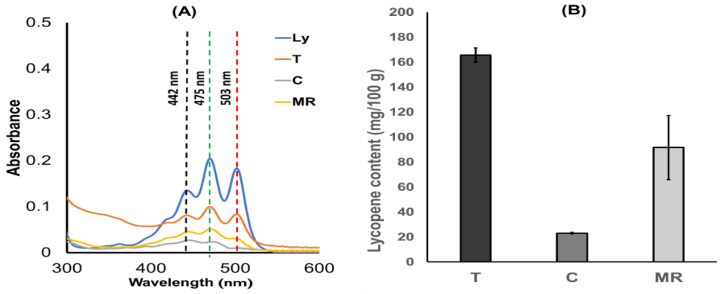
(**A**) UV-VIS spectra of standard lycopene (Ly), tomato (T), carrot (C), mixed red vegetable (MR), and (**B**) lycopene content in tomato (T), carrot (C), and mixed red vegetable (MR).

**Figure 2 ijms-25-11717-f002:**
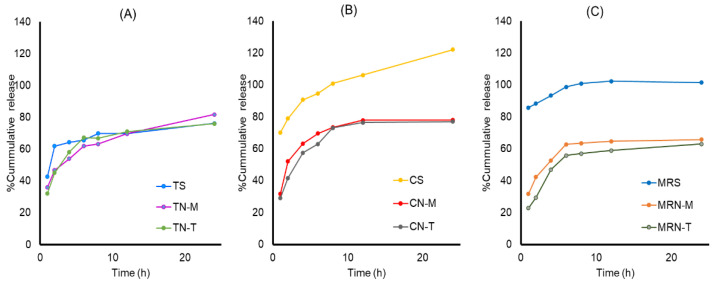
Comparison of lycopene-release profiles from (**A**) tomato extract solution (TS), niosome-entrapped tomato extract prepared by MF method (TN−M) and THF method (TN−T); (**B**) carrot extract solution (CS), niosome-entrapped carrot extract prepared by MF method (CN−M) and THF method (CN−T); (**C**) mixed red vegetable solution (MRS), niosome-entrapped mixed red vegetable prepared by MF method (MRN−M) and THF method (MRN−T).

**Figure 3 ijms-25-11717-f003:**
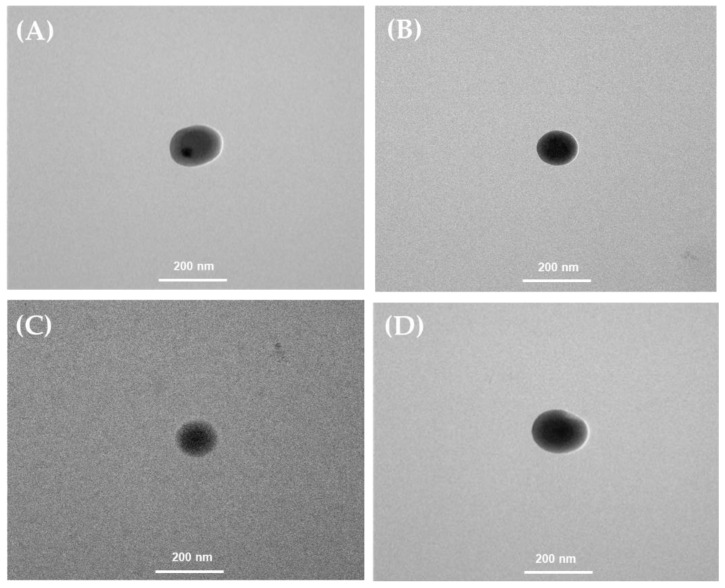
The morphology of niosome formulations prepared by the THF method imaged (magnification ×25,000) using transmission electron microscopy (TEM): (**A**) blank niosome formulation (BN); (**B**) niosome-entrapped tomato extract (TN); (**C**) niosome-entrapped carrot extract (CN); (**D**) niosome-entrapped mixed red vegetable extract (MRN).

**Figure 4 ijms-25-11717-f004:**
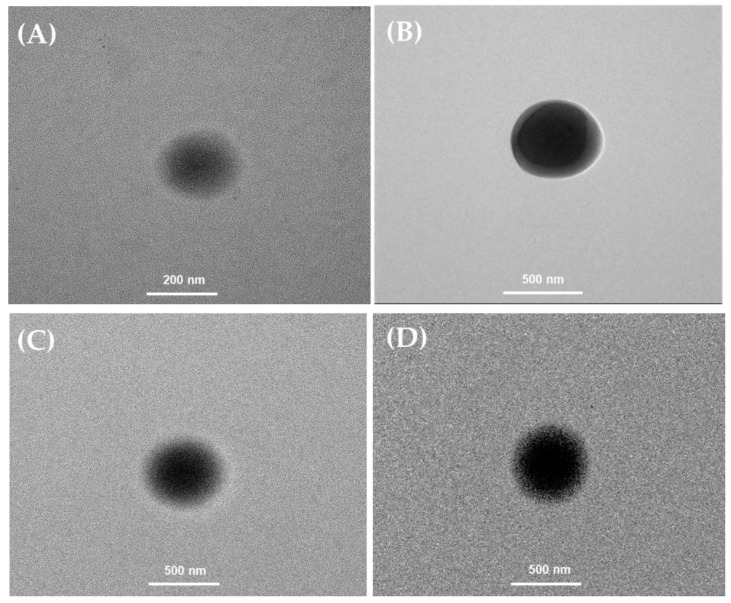
The morphology of niosome formulations prepared by MF method imaged (magnification ×25,000) using transmission electron microscopy (TEM): (**A**) blank niosome formulation (BN); (**B**) niosome-entrapped tomato extract (TN); (**C**) niosome-entrapped carrot extract (CN); (**D**) niosome-entrapped mixed red vegetable extract (MRN).

**Figure 5 ijms-25-11717-f005:**
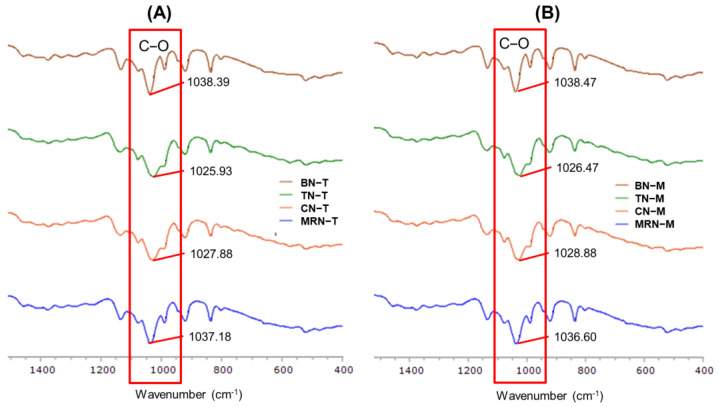
FT-IR spectra of (**A**) niosomes prepared by the thin-film hydration method: BN−T, TN−T, CN−T, MRN−T; and (**B**) niosome prepared by the microfluidic method: BN−M, TN−M, CN−M, MRN−M.

**Figure 6 ijms-25-11717-f006:**
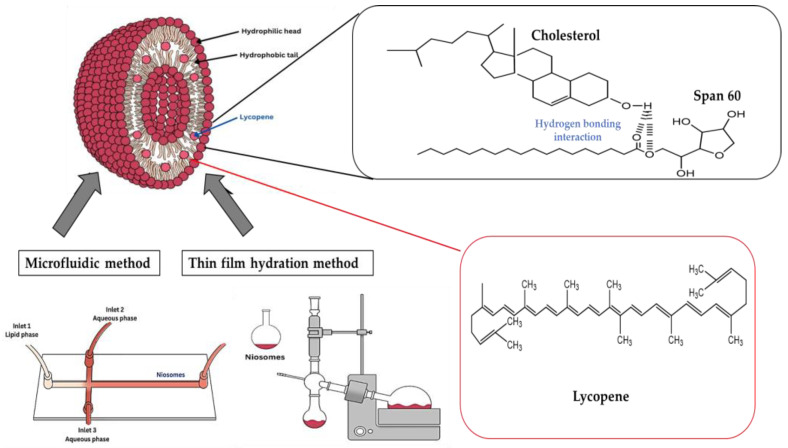
Illustration of the interaction between lycopene and Span 60 and cholesterol in niosome formations prepared by the microfluidic method and the thin-film hydration method.

**Figure 7 ijms-25-11717-f007:**
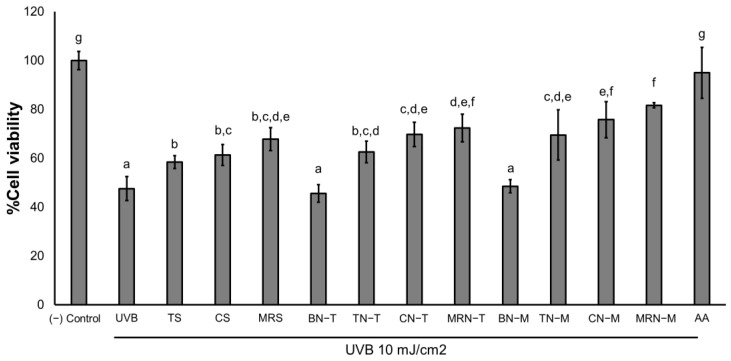
Viability of HaCaT cells exposed to UVB radiation (UVB) and pretreated with tomato (TS), carrot (CS), and mixed red vegetable extracts (MRS), niosome formulations prepared by thin-film hydration method (BN−T, TN−T, CN−T, MRN−T), niosome formulations prepared by microfluidic method (BN−M, TN−M, CN−M, MRN−M), and 50 µg/mL ascorbic acid (AA). Data represent the mean ± SD values of three replicates. Significance was evaluated by the Duncan’s test (*p* < 0.05). Letters a–g indicate significant differences between the groups.

**Figure 8 ijms-25-11717-f008:**
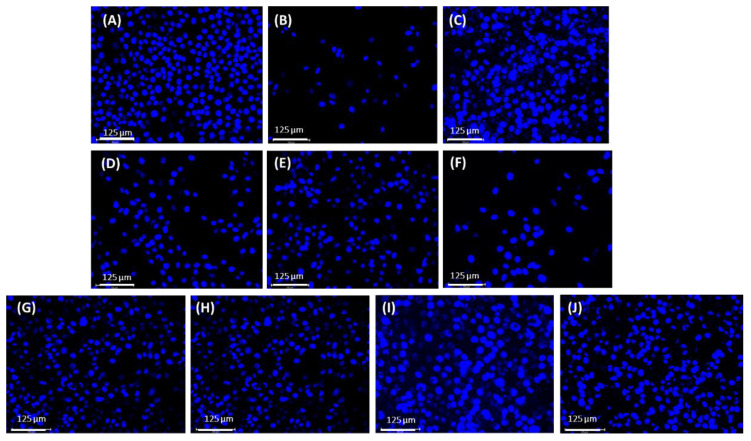
Morphology of HaCaT cells using fluorescence (DAPI) staining: non-treated cells (**A**), cells exposed to 10 mJ/cm^2^ UVB radiation (**B**), and UVB radiation of HaCaT cells after 24 h of pretreatment with ascorbic acid 50 µg/mL (**C**), tomato (**D**), carrot (**E**), mixed red vegetables (**F**), BN−M (**G**), TN−M (**H**), CN−M (**I**), MRN−M (**J**).

**Figure 9 ijms-25-11717-f009:**
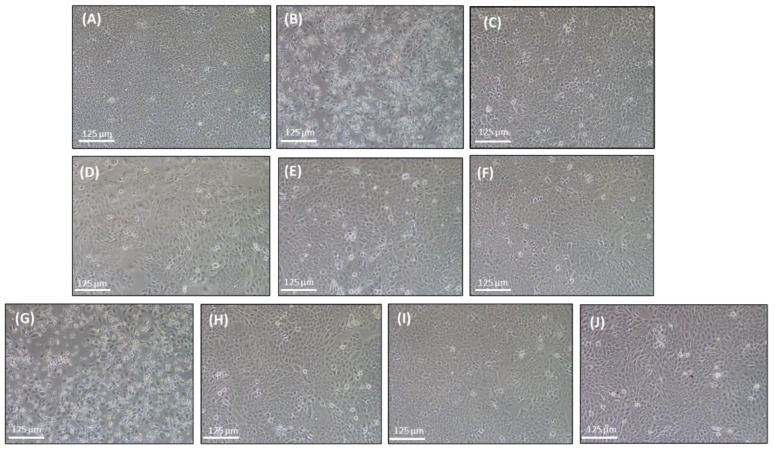
Morphology of HaCaT cells: non-treated cells (**A**), cells exposed to 10 mJ/cm^2^ UVB radiation (**B**), and UVB radiation after 24 h of pretreatment with ascorbic acid 50 µg/mL (**C**), tomato (**D**), carrot (**E**), mixed red vegetables (**F**), BN−M (**G**), TN−M (**H**), CN−M (**I**), MRN−M (**J**).

**Figure 10 ijms-25-11717-f010:**
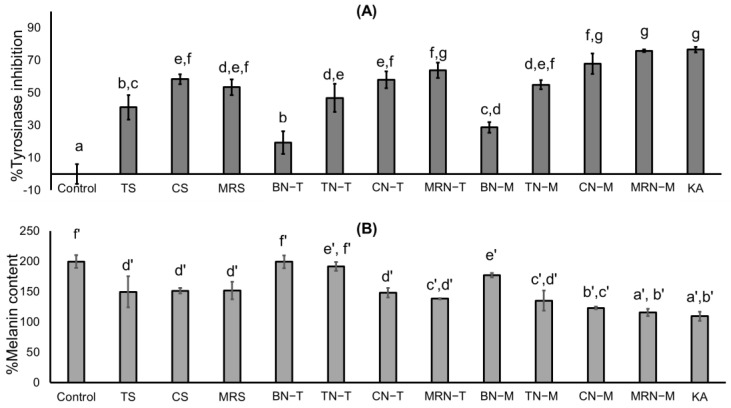
Effect of tomato (TS), carrot (CS), and mixed red vegetable extracts (MRS), niosome formulations (blank niosome (BN−T), TN−T, blank−niosome (BN−M), TN−T, CN−T, MRN−T, blank niosome (BN−M)), TN−M, CN−M, MRN−M) and 100 µg/mL kojic acid (KA) on B16F10 melanoma cells stimulated by α-melanocyte-stimulating hormone (α-MSH): tyrosinase inhibition (**A**) and melanin content (**B**). Control was B16F10 melanoma cells stimulated by α-MSH without sample treatment. Data represent the mean ± SD values of three replicates. Significance was evaluated by Duncan’s test (*p* < 0.05), letters a–g indicate significant differences in tyrosinase inhibition between the groups, and letters a′–f′ indicate significant differences in melanin content between the groups.

**Figure 11 ijms-25-11717-f011:**
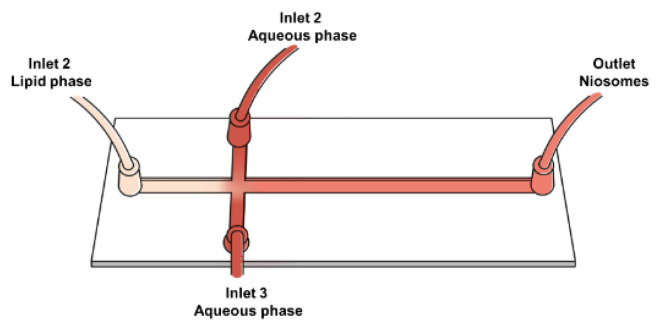
Microfluidic platform for the cross-junction microchannel for lycopene carriers, a schematic of a hydrodynamic flow-focusing (HFF) platform. Inlet 1 = Lipid phase (Span 60, cholesterol), Inlets 2 and 3 = Aqueous phase-contained extracts (tomato, carrot, or mixed red vegetables).

**Table 1 ijms-25-11717-t001:** Entrapment efficiency (EE, %), Zeta potential (mV), Particle size (Size, nm), and PDI values of niosome formulations prepared by thin-film hydration (THF) and microfluidic (MF) methods.

Method	Formulation	EE (%)	Zeta Potential (mV)	Size (nm)	PDI
Thin-film (THF)	BN−T	−	−44.80 ± 1.64	245.17 ± 43.98	0.54 ± 0.01
	TN−T	90.59 ± 7.09	−36.76 ± 1.97	477.00 ± 56.21	0.78 ± 0.21
	CN−T	92.02 ± 0.84	−33.13 ± 2.76	461.93 ± 55.91	0.72 ± 0.14
	MRN−T	95.69 ± 3.76	−31.57 ± 0.25	457.47 ± 50.06	0.70 ± 0.07
Microfluidic (MF)	BN−M	−	−61.50 ± 0.90 *	278.40 ± 18.47	0.40 ± 0.04 *
	TN−M	92.55 ± 0.16	−48.77 ± 3.54 *	281.73 ± 16.40 *	0.29 ± 0.03 *
	CN−M	97.22 ± 1.26	−42.53 ± 1.19 *	237.97 ± 42.54 *	0.22 ± 0.01 *
	MRN−M	90.37 ± 0.54*	−31.33 ± 1.11	256.83 ± 5.16 *	0.58 ± 0.19

Data represent the mean ± SD values of three replicates. Statistical significance was evaluated by paired *t*-test (* *p* < 0.05 compared with THF method for the same extract). The formulations prepared by the THF method are blank niosomes (BN−T) and niosomes of tomato (TN−T), carrot (CN–T), and mixed red vegetable (MRN−T) extracts. The formulations prepared by the MF method are blank niosomes (BN−M) and niosomes of tomato (TN−M), carrot (CN−M), and mixed red vegetable (MRN−M) extracts.

**Table 2 ijms-25-11717-t002:** Stability test for determination of entrapment efficiency (%), Zeta potential (mV), particle size (nm), and PDI values of niosome formulations made by thin-film hydration and microfluidic methods before and after storage.

Formulation	EE (%)		Zeta Potential (mV)		Size (nm)		PDI	
	Before	After	Before	After	Before	After	Before	After
BN−T	−	−	−44.80 ± 1.64	−40.20 ± 1.15	245.17 ± 43.98	217.67 ± 23.80	0.54 ± 0.01	0.50 ± 0.12
TN−T	90.59 ± 7.09	75.25 ± 1.09 *	−36.76 ± 1.97	−34.73 ± 3.61	477.00 ± 56.21	403.57 ± 30.17	0.78 ± 0.21	0.70 ± 0.27
CN−T	92.02 ± 0.84	82.10 ± 0.29 *	−33.13 ± 2.76	−30.47 ± 1.10	461.93 ± 55.91	482.27 ± 15.29	0.72 ± 0.14	0.71 ± 0.06
MRN−T	95.69 ± 3.76	82.21 ± 1.59 *	−31.57 ± 0.25	−29.80 ± 0.60 *	457.47 ± 50.06	447.90 ± 15.29	0.70 ± 0.07	0.58 ± 0.07
BN−M	−	−	−61.50 ± 0.90	−58.50 ± 0.60	278.40 ± 18.47	277.70 ± 16.13	0.40 ± 0.04	0.41 ± 0.05
TN−M	92.55 ± 0.16	82.27 ± 3.85 *	−48.77 ± 3.54	−44.90 ± 10.52	281.73 ± 16.40	263.47 ± 36.26	0.29 ± 0.03	0.26 ± 0.07
CN−M	97.22 ± 1.26	83.49 ± 4.68 *	−42.53 ± 1.19	−44.30 ± 4.97	237.97 ± 42.54	238.40 ± 35.28	0.22 ± 0.01	0.31 ± 0.06
MRN−M	90.37 ± 0.54	77.13 ± 11.73	−31.33 ± 1.11	−41.93 ± 2.16	256.83 ± 5.16	223.87 ± 14.26	0.58 ± 0.19	0.31 ± 0.06

Data represent the mean ± SD values of three replicates. Significant differences were compared between storage conditions by a paired sample *t*-test (* *p* < 0.05).

**Table 3 ijms-25-11717-t003:** Composition of non-ionic surfactants, cholesterol, and extract loading in niosome formulations (T = tomato, C = Carrot, MR = mixed red vegetables) prepared by thin-film hydration (THF) method.

Formulation	Ratio (mg)	Hydration Volume (mL)	Molar Ratio (%)	Total Volume (mL)
BN−T Span 60/Cholesterol	12.90:11.40	10.00	50:50	10.00
TN−T Span 60/Cholesterol/T	12.90:11.40:100	10.00	50:50	10.00
CN−T Span 60/Cholesterol/C	12.90:11.40:100	10.00	50:50	10.00
MRN−T Span 60/Cholesterol/MR	12.90:11.40:100	10.00	50:50	10.00

The formulations prepared by the THF method are BN−T, TN−T, CN−T, and MRN−T (blank niosome and niosomes of tomato, carrot, and mixed red vegetables, respectively).

## Data Availability

Data are contained within the article and Appendix A.

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
