# Peer review of "Comparative Study of Lycopene-Loaded Niosomes Prepared by Microfluidic and Thin-Film Hydration Techniques for UVB Protection and Anti-Hyperpigmentation Activity"

_ijms, 2024, doi:10.3390/ijms252111717_

Round 1
Reviewer 1 Report
Comments and Suggestions for Authors
1. Include HPLC analysis or GCMS analysis for confirmation of lycopene in vegetables
2. Include dual dye fluorescent assay for cells morphology
3. Include the scale bar for all images in Figure 10
4. The TEM micrographs confirm that the size of the neosomes is 200 to 500 nm. So, how it works. Explain?
5. Include the zeta potential graphs
6. Include the X-Axis for FTIR
7. Check throughout the manuscript. Many errors are noticed. For example, In 2.4.8. FTIR is mistakenly mentioned in FITR; in vitro need to change in italics
8. Reduce the plaquarism to below 20%
9. Include more recent references
10. Include photographs of extracts of lycopene in vegetables
Comments on the Quality of English Language
Extensive editing of the English language is required.
Reviewer 2 Report
Comments and Suggestions for Authors
The paper “Comparative Study of Lycopene-Loaded Niosomes Prepared by Microfluidic and Thin Film Hydration Techniques for UVB Protection and Anti-Hyperpigmentation Activity” investigates the impact of different preparation methods on the characterization of lycopene extract niosomes sourced from vegetables, as well as their biological activities. The results indicated that lycopene-niosome prepared by MF exhibited high uniformity, and homogeneity (with low PDI value), and maintained smaller sizes when processed through a chip utilizing a hydrodynamic flow focusing (HFF) plat-form, compared to conventional thin-film hydration method (THF).
The paper is interesting, the methodology is adequate and explicitly stated and the subject is very topical. The results and conclusions are remarkable and for this reason, I recommend the publication of this study after a minor revision.
Therefore, the authors are invited to clarify the following aspects:
· The manuscript should be checked for the possible writing errors.
· Abstract: The sentence: The lycopene-loaded niosome preparation by the microfluidic method (MF) was com-pared with the conventional thin-film hydration method (THF) was studied UVB protection and Anti-Hyperpigmentation activities.” is not clear
· The novelty aspect is missing from the abstract and needs to be clearly articulated. What parts do you consider original or relevant for the field? What specific gap in the field does the paper address?
· For Materials and methods section: for each equipment and device used must be identified through the model, trade mark and country of origin.
· I would suggest that in the conclusions include some final considerations on the novelties that this work provides with respect to others already existing in the bibliography.
Overall, this work is of scientific interest and is relevant within its scientific area.
Author Response
Please ser the attachment.
